# Crosstalk between Nutrition, Insulin, Juvenile Hormone, and Ecdysteroid Signaling in the Classical Insect Model, *Rhodnius prolixus*

**DOI:** 10.3390/ijms24010007

**Published:** 2022-12-20

**Authors:** Jimena Leyria, Samiha Benrabaa, Marcela Nouzova, Fernando G. Noriega, Lilian Valadares Tose, Francisco Fernandez-Lima, Ian Orchard, Angela B. Lange

**Affiliations:** 1Department of Biology, University of Toronto Mississauga, Mississauga, ON L5L 1C6, Canada; 2Biology Center of the Academy of Sciences of the Czech Republic, Institute of Parasitology, 37005 České Budějovice, Czech Republic; 3Department of Biological Sciences and Biomolecular Science Institute, Florida International University, Miami, FL 33199, USA; 4Department of Parasitology, University of South Bohemia, 37005 České Budějovice, Czech Republic; 5Department of Chemistry and Biochemistry and Biomolecular Science Institute, Florida International University, Miami, FL 33199, USA

**Keywords:** insect, corpus allatum, ovary, hormone titers, endocrine signaling

## Abstract

The rigorous balance of endocrine signals that control insect reproductive physiology is crucial for the success of egg production. *Rhodnius prolixus*, a blood-feeding insect and main vector of Chagas disease, has been used over the last century as a model to unravel aspects of insect metabolism and physiology. Our recent work has shown that nutrition, insulin signaling, and two main types of insect lipophilic hormones, juvenile hormone (JH) and ecdysteroids, are essential for successful reproduction in *R. prolixus*; however, the interplay behind these endocrine signals has not been established. We used a combination of hormone treatments, gene expression analyses, hormone measurements, and ex vivo experiments using the corpus allatum or the ovary, to investigate how the interaction of these endocrine signals might define the hormone environment for egg production. The results show that after a blood meal, circulating JH levels increase, a process mainly driven through insulin and allatoregulatory neuropeptides. In turn, JH feeds back to provide some control over its own biosynthesis by regulating the expression of critical biosynthetic enzymes in the corpus allatum. Interestingly, insulin also stimulates the synthesis and release of ecdysteroids from the ovary. This study highlights the complex network of endocrine signals that, together, coordinate a successful reproductive cycle.

## 1. Introduction

In all organisms, reproduction controls the dispersion of individuals and populations. In the case of insects, this involves not only the spread of beneficial species but also that of pests and vectors of diseases. There exists an urgent need for the development of novel, biorational insecticides, that are species-specific and with less negative impact on the environment. Due to their potency and potential for less environmental impact compared to chemical pesticides, synthetic hormones have been widely used for pest control; however, greater efforts need to be made to understand the nature of hormonal regulation in every insect species at the molecular level in order to produce effective vector control [1,2,3,4,5]. Thus, the study of the hormonal control of insect reproduction is highly relevant from an economic, ecological and epidemiological point of view. Reproductive processes have been extensively investigated in several insect species, and although each species may have its own particular features, general assertions can often be made. 

For example, vitellogenesis is energetically costly for insect females, is essential for egg production, and involves the synthesis of a huge amount of yolk protein precursors (YPPs) primary by the fat body, a multifunctional organ analogous to vertebrate adipose tissue and liver. Vitellogenin (Vg) is then released into the hemolymph and internalized by the maturing oocytes [2,3]. The energy available for allocation to vitellogenesis is limited and mainly obtained from nutrients acquired from meals [4]. 

The regulation of insect reproductive capacity involves the release of chemical messengers such as peptides or lipophilic hormones that can act in two ways; directly coordinating a physiological event or influencing the production of other messengers that do [2,5]. The insulin/insulin-like peptides (ILPs)/insulin growth factors (IGFs)/relaxin superfamily comprises several widely studied hormones, whose signaling pathways, including receptor proteins and their downstream participating elements, are highly conserved throughout the Metazoa [6,7]. The number and type of insect ILPs varies between species; for example, two ILPs and one IGF were identified in the migratory locust, *Locusta migratoria* [8], while eight were found in the fruit fly, *Drosophila melanogaster* [9], thirty-eight in the domestic silk moth, *Bombyx mori* [10] and four ILPs and one IGF in the blood-sucking insect, *Rhodnius prolixus* [11,12,13]; all of them regulate different physiological events either alone or in combination, and may possess functional redundancy and/or mediate feedback regulation [6]. In insects, ILPs are mainly synthesized and released from the brain, although several ILPs have also been reported to be produced by other tissues, including the fat body and ovaries [14,15,16,17,18,19]. As a response to a positive nutritional state, insulin signaling stimulates vitellogenesis promoting egg production [2,3,14], or ovulation leading to egg laying [18,19]. With regard to vitellogenesis, ILPs not only stimulate the synthesis of the main YPP, Vg, but also control biosynthesis and secretion of lipophilic hormones [17,20,21,22,23,24,25,26]. 

In insects, there are two main types of lipophilic hormones, namely ecdysteroids and juvenile hormones (JHs), that are well known for their critical roles during development and metamorphosis [27]. These hormones are also synthesized in adult insect females, promoting mainly vitellogenesis but also ovulation or oviposition, depending on the insect species. In adult females, the ovaries are the main source of synthesis and release of ecdysteroids, cholesterol-derived insect hormones. Insects convert cholesterol into ecdysone and 20-hydroxyecdysone (20E), physiologically the most relevant ecdysteroid, by hydroxylation and oxidation reactions performed by cytochrome P450 enzymes encoded by Halloween genes [28]. Ecdysteroids have autocrine and paracrine regulatory roles in the ovary since they can control ovarian growth, oocyte maturation, follicle cell development and choriogenesis [29,30,31,32,33,34,35,36,37]. Moreover, an endocrine action has been widely studied not only in dipterans, but also in hymenopterans and lepidopterans, where circulating ecdysteroids act upon the fat body to stimulate full Vg production [5,38,39,40]. Additionally, in some insects, circulating ecdysteroids are involved in controlling ovulation and possibly oviposition [34,41,42].

JHs encompass a family of acyclic sesquiterpenoids, produced and released into the hemolymph by the corpus allatum (CA), an endocrine gland with neural connections to the brain and closely associated with another neuroendocrine organ, the corpus cardiacum (CC). A total of eight JH homologs have been identified, which include JH 0, JH I, JH II, JH III, 4-methyl JH I (Iso-JH 0), JHB_3_, JHSB_3_ and the non-epoxidated methyl farnesoate [43]. The biosynthetic pathway (Figure 1A) involves the same 13 enzymatic reactions and is conventionally divided into early (mevalonate pathway) and late (JH-branch) steps; only changes in substrate specificity seems to be responsible for some of the structural differences among JHs [43,44]. JH production in the CA is mediated by two important classes of allatoregulatory neuropeptides, which bind to specific receptors to modulate JH synthesis by stimulating (allatotropins/ATs) or inhibiting (allatostatins/ASTs) the CA [45]. In most of the Orthoptera, Blattodea and Hemiptera orders, JH is essential for full egg production [3], inducing Vg synthesis in the fat body and promoting the opening of intercellular spaces in the follicular epithelium (patency) which facilitate the selective uptake of circulating YPPs by oocytes [3]. Interestingly, in *D*. *melanogaster* adult females, JH regulates not only Vg uptake but also maintains egg shape and induces ovulation [46].

The blood-sucking insect *R. prolixus* has been a model for studying reproduction in insects for the last 100 years [41,47]; indeed, the concept of a “metamorphosis inhibitory hormone” named JH, and later reported to be involved in insect reproduction, was first described by Sir Vincent B. Wigglesworth using *R*. *prolixus* females [48]. *R*. *prolixus* also has relevance in public health since it is a vector of the parasite *Trypanosoma cruzi*, the etiological agent of Chagas disease, an illness endemic in Latin America but that currently is becoming a complex global health problem [49]. Recently, we demonstrated that insulin signaling is activated after a blood meal to modulate the vitellogenic process [16,17]. Additionally, we have elucidated mechanisms and target genes underlying JHSB_3_ action, the naturally occurring JH homolog in *R*. *prolixus* [50], identifying the genes Tai and Kr-h1, and reporting the physiological roles of Met-Tai-Kr-h1 in the JH signaling cascade for successful reproduction [51]. Despite this very important advance, JHSB_3_ titers in hemolymph after a blood meal in adult *R*. *prolixus* have not been reported. Moreover, neither AST nor AT have been tested for activity on JH biosynthesis in *R*. *prolixus*. A role of ecdysteroids in *R*. *prolixus* ovulation and possibly oviposition is well described [41], but more recently we also found that ecdysteroids regulate ovarian growth and oocyte maturation [52]. However, the crosstalk between nutrition, insulin, ecdysteroids and JHSB*_3_* signaling pathways regulating vitellogenesis is not yet known in this historic insect model *R*. *prolixus*. Using a combination of hormone treatments, gene expression analyses, hormone measurements, and ex vivo experiments, we find that following a blood meal, circulating JH levels increase, a process mainly driven through insulin and allatoregulatory neuropeptides. In turn, JH feeds back to provide some control over its own biosynthesis by regulating the expression of critical biosynthetic enzymes in the corpus allatum. Interestingly, insulin also stimulates the synthesis and release of ecdysteroids from the ovary. Overall, we show for the first time the interplay between nutrition and the endocrine system that defines the hormonal environment for successful reproduction in *R*. *prolixus* females. 

## 2. Results

### 2.1. Transcript Levels of JHSB_3_ Biosynthetic Enzymes and Circulating Levels of JHSB_3_ Depend on Nutritional State 

The JH biosynthetic pathway includes 13 enzymatic reactions generally divided into early and late steps (Figure 1A). In R. prolixus, the sequences of the 13 putative genes encoding for those enzymes have been reported [50]. Since the CA is the main site of JH production, the impact of nutritional state on mRNA expression of biosynthetic enzymes was studied in the CC-CA complex (Figure 1B). The transcript levels for 5 of 8 enzymes from the early step, acetyl-CoA-thiolase (thiol), HMG-CoA reductase (HMGR); diphosphomevalonate decarboxylate (PP-MEVD); isopentenyl diphosphate isomerase (IPPI); farnesyl diphosphate synthase (FPPS), show a statistically significant increase 24 h post blood meal. In the late step, mRNA expression for 4 of 5 enzymes, farnesyl diphosphate pyrophosphatase (FPPP), farnesol dehydrogenase (FALDH); juvenile hormone acid methyltransferase (JHAMT), and methyl farneseoate epoxidase (Epox), are also upregulated 24 h post blood meal. Transcripts for the two last enzymes, JHAMT and Epox, are the most abundant, where JHAMT levels are 3-fold higher with respect to Epox levels and more that 25-fold higher than the other 11 enzymes. However, JHAMT expression still increases 2-fold post blood meal with respect to unfed insects, while Epox levels increase around 5-fold. The JHAMT enzyme has been proposed as the rate-limiting enzyme in JH biosynthesis, and its mRNA level and enzyme activity are closely correlated with the patterns of JH synthesis [53]. Thus, to evaluate this correlation in R. prolixus, we measured the JH titers in the hemolymph of adult females during three time points of the unfed condition and throughout 10 days post blood meal. We first performed a screening for five different epoxidated JH homologs, JH I, JH II, JH III, JHB_3_, JHSB_3_, and the non-epoxidated MF, using LC-MS/MS. The results revealed only the presence of JHSB_3_ in the hemolymph at all the nutritional conditions analyzed (Appendix A). JHSB_3_ hemolymph titers decrease in unfed adult females as the days after ecdysis progress, with the higher values at 2–3 days after ecdysis (~0.2 fmol/µL of hemolymph) and the lower after 30–35 days (~0.04 fmol/µL of hemolymph) (Figure 1C). Circulating JHSB_3_ levels are higher after a blood meal, the main stimulus for vitellogenesis and egg growth (Figure 1D), with a significant increase 24 h after a blood meal (~4 fmol/µL of hemolymph) with respect to unfed insects (10–12 days post ecdysis, ~0.095 fmol/µL of hemolymph; *p* < 0.01). The values found after feeding are relatively constant until 6 days post blood meal, a time when egg laying begins [51], and then decrease, with lower titers for the remaining days (between 1.8 and 0.9 fmol/µL of hemolymph) but still around 10-fold higher than those found in unfed females (Figure 1D). 

### 2.2. Insulin Signaling Regulates Transcript Expression of JH Biosynthetic Enzymes

To test whether the insulin signaling pathway is able to induce JH production, we first treated newly emerged adult females (when JHSB_3_ titers are decreasing) with exogenous insulin, and measured mRNA levels of the JH biosynthetic enzymes in the CC-CA complex 24 h after treatment (Figure 2A). The results show an increase in mRNA expression of four enzymes of the early step, thiol, HMG-CoA synthase (HMGS), phosphomevalonate kinase (P-MEVK) and FPPS, and four enzymes of the late step, FPPP, farnesol dehydrogenase (FOLD), JHAMT and Epox (Figure 2A). The highest change between insulin-treated insects and controls is observed for Epox expression (~3-fold higher than control insects, *p* < 0.01). We also quantified Rhopr-Vg1 mRNA in the fat body and circulating Vg in the same insects where the enzyme transcripts were quantified. The results show an increase of Rhopr-Vg1 mRNA expression in the fat body (Figure 2B, *p* < 0.05) as well as an increase in vitellogenin in the hemolymph after insulin injection (Figure 2C). Interestingly, when JHSB_3_ hemolymph titers are assayed in unfed insulin-injected insects, a non-statistically significant increase is observed (Figure 2D). To add support for insulin signaling working directly on the CC-CA complex, we evaluated transcript levels of the two insulin receptors present in R. prolixus, Rhopr-IR1 and Rhopr-IR2 [17]. The CC-CA complex has relatively higher expression of Rhopr-IR2 transcript than Rhopr-IR1 in both unfed and fed insects. Following a blood meal, transcript levels of Rhopr-IR1 and Rhopr-IR2 decrease (Figure 3), a pattern already shown for these receptors in fat body and ovaries of R. prolixus [16,17]. 

As an in vivo stimulation is systemic, an indirect effect of insulin on the expression of JH biosynthetic enzymes cannot be ruled out; thus, we confirmed the direct effect of insulin on the CC-CA complexes by ex vivo assays. When the isolated CC-CA complex is incubated in a medium containing insulin, an upregulation of mRNA levels for 10 of the 13 biosynthetic enzymes is observed (Figure 4A). Detecting significant changes in JHSB_3_ incubation medium levels is challenging since the hormonal concentration is very low. In our ex vivo experiments, therefore, we measured the rate of JH synthesis of groups of three CC-CA complexes incubated for a period of 4 h, so JHSB_3_ can accumulate in larger amounts in the incubation medium and thereby the effect of insulin could be more easily detected. JHSB_3_ levels in the incubation medium after insulin treatment increase significantly with respect to controls (*p* < 0.01) (Figure 4B). A classical protein activated downstream of the conserved insulin signaling is PI3-kinase (PI3K); therefore, we incubated the CC-CA complexes with the PI3K inhibitor LY294002 prior to insulin treatment. The results show that JHSB_3_ levels in the incubation medium are reduced when LY294002 is present, even with the addition of insulin (Figure 4B). In order to evaluate the possible influence of glucose in the regulation of JH production, we performed ex vivo experiments using the CC-CA complexes connected to the brains (CC-CA-Brain) in a medium containing glucose. The results show a statistically significant increase of the JHSB_3_ accumulated in the incubation medium containing glucose with respect to control (CC-CA-Brain) (*p* < 0.05) (Figure 4C). 

### 2.3. JHSB_3_ Appears to Regulate Its Own Biosynthesis in the Corpus Allatum 

Recently, we introduced a possible new form of JH regulation in R. prolixus, whereby the Met-Tai complex (the JH receptor and its partner) might act as a hormonal sensor in the CA, promoting a negative feedback loop that keeps JHSB_3_ biosynthesis and CA activity under control [51]. To test whether JHSB_3_ controls its own biosynthesis, we treated newly emerged adult females with exogenous JHSB_3_ and 24 h later measured mRNA expression in the CC-CA complexes (Figure 5). Transcript levels of JHAMT (*p* < 0.05) and Epox (*p* < 0.01), the two biosynthetic enzymes quantitatively more important in the late step, are downregulated in the CC-CA complexes from JHSB_3_-treated females. 

### 2.4. JHSB_3_ Synthesis Is Regulated by Allatotropin and Allatostatin

Different factors have been shown to regulate JH biosynthesis in insects, among them, ATs and ASTs via their G protein-coupled receptors (GPCRs) [45]. ATs are one family of peptides whereas for the ASTs there are three distinct families in insects, FGLa/ASTs (or A-type ASTs), MIP/ASTs (or B-type ASTs) and PISC/ASTs (or C-type ASTs) [45]. Here, we tested the effects of AT and one FGLa/AST on JH levels in ex vivo assays using the CC-CA complexes from unfed or fed females, as indicated. The results show a stimulatory effect of AT on JH biosynthesis at 10^−7^ M (*p* > 0.05; Figure 6A) and an inhibitory effect by FGLa/AST at 10^−7^ M (Figure 6B), but again not significant (*p* > 0.05). Furthermore, when mRNA expression of GPCRs is tested in different nutritional states (Figure 6C), our results show that the mRNA expression of AT-R is upregulated after a blood meal (Figure 6D, *p* < 0.05), when the JH hemolymph titers increase. The possible involvement of FGLa/ASTs in the direct inhibition of JH biosynthesis is also supported by the presence of ASTA-R mRNA in the CC-CA complex, whose levels are not modified by the nutritional state (Figure 6E). 

### 2.5. Insulin Signaling Regulates Transcript Expression of Ecdysteroid Biosynthetic Enzymes

Among seven enzymes that are involved in the biosynthesis of ecdysteroids, five of them are cytochrome P450 mono-oxygenases (CYPs). The genes for these seven enzymes, collectively known as Halloween genes, include *neverland* (nvd), *non-molting glossy/shroud* (sro), *spook* (spo), *phantom* (phm), *disembodied* (dib), *shadow* (sad) and *shade* (shd) (Figure 7A). Recently, the Halloween genes were reported in *R. prolixus* [52]. Here, we evaluated the potential involvement of insulin in regulating ecdysteroid synthesis in the ovaries of *R. prolixus* females. We show that insulin injection into unfed females, when ecdysteroid hemolymph titers are low [52], induces an upregulation of transcript expression of *Phm* (*p* < 0.05), *Sad* (*p* < 0.01) and *Shd* (*p* < 0.05) (Figure 7B1). When ecdysteroid hemolymph titers are measured in insulin-injected females, a statistically significant increase with respect to control insects is observed (*p* < 0.05) (Figure 7B2). In order to confirm this interplay, we performed ex vivo experiments, using ovaries from unfed insects. The results show a statistically significant increase of all the Halloween genes, *Nvd* (*p* < 0.05), *Spo* (*p* < 0.05), *Phm* (*p* < 0.05), *Dib* (*p* < 0.001), *Sad* (*p* < 0.01) and *Shd* (*p* < 0.05, *n* = 7*–*8) (Figure 7C1), in ovaries incubated with insulin compared to controls. The ecdysteroids levels in the incubation medium containing insulin are significantly higher with respect to controls (*p* < 0.01), (Figure 7C2). 

## 3. Discussion

The rigorous balance of endocrine signals is vital for insect reproductive physiology, especially in females. Processes such as courtship, mating, oogenesis, ovulation, fertilization, and oviposition demand energy and endocrine regulation, and a hormone imbalance would be devastating for these processes. This work illustrates a network between important endocrine factors leading to successful reproduction in *R. prolixus,* the blood gorging model species upon which the foundations of insect physiology, endocrinology, and development were built. 

### 3.1. JHSB_3_ Biosynthetic Enzyme Activities and Circulating Levels of JHSB_3_ Are Closely Correlated and Depend on Nutritional State 

As found in other insect species, the hormonal and neural control of reproductive processes in *R. prolixus* includes synchronization of signals from the midgut, central nervous system, CC-CA complex, fat body, and reproductive tissues [47]. Wang and Davey [54] reported for the first time that JH is the main gonadotrophic hormone stimulating YPP synthesis in the fat body of triatomine insects. Recently, we extended that observation and found that JHSB_3_ promotes Vg production, not only in the fat body, but also in the ovaries [51]. JH is synthesized in a complex and unique biosynthetic manner according to the insect order and physiological needs; different JH homologs can be present in different combinations for each insect. Here, we report that, as was shown in nymphs [50], JHSB_3_ is the only JH molecular form present in adult female *R. prolixus*, as described in a related species, *Dipetalogaster maxima* [55], and in several heteropterans [56]. Additionally, we show for the first time the JHSB_3_ levels in the hemolymph throughout the reproductive cycle of *R. prolixus* females. The hemolymph titers of JH increase after a blood meal and do so in concert with an elevation in the transcripts for the biosynthetic enzymes in the CC-CA, confirming that enzyme activity is closely correlated with the patterns of JH synthesis and release [53]. It is also interesting to note that the enzymes involved in the early steps of the mevalonate pathway are also responsible for the production of other terpenoids, such as defensive secretions and pheromones; thus, some variation in their expression related to these other activities might be expected [44]. The major contribution to JH synthesis appears to be from the final two enzymes of the JH-branch, JHAMT and Epox. Epoxidation of JH was recently proposed as a key to improving insect reproductive fitness [57], and interestingly in *R. prolixus*, *Epox* has the largest change in mRNA levels after a blood meal confirming its importance. 

### 3.2. Insulin Signaling Regulates the Expression of JH Biosynthetic Enzymes

Although autogeny, the capacity of producing an initial batch of eggs before obtaining a blood meal, has been reported in *R. prolixus* [58], blood gorging is required for complete vitellogenesis and egg production [41]. The managing of this nutritional input is mainly attributed to the insulin signaling pathway, which has been extensively studied in different insect species and shown to positively impact vitellogenesis and oocyte growth [2,14,59]. In *R. prolixus, Vg* mRNA expression in the fat body is also modulated by insulin [14]. In the yellow fever mosquito *Aedes aegypti*, the German cockroach *Blattella germanica*, the red flour beetle *Tribolium castaneum,* and in the desert locust, *Schistocerca gregaria*, a stimulatory effect of insulin, alone or in conjunction with lipophilic hormones, on Vg synthesis by the fat body has also been demonstrated [23,24,26,32]. In *D. melanogaster, B. germanica*, *A. aegypti,* the American cockroach *Periplaneta americana* and the tobacco cutworm, *Spodoptera litura,* insulin signaling regulates JH synthesis in the CA [25,26,60,61,62]. Here, we show that insulin also acts on the CA of *R. prolixus* females promoting not only the synthesis but also the release of JHSB_3_; in agreement with this finding, the transcripts for both insulin receptors, *Rhopr-IR1* and *Rhopr-IR2*, are expressed in the CC-CA complex. Recently, we also reported that ILPs are stored in the medial neurosecretory cells (MNSCs) of the brain [17,63]. Vafopoulou and Steel [64] also found MNSC axons stained with ILPs projecting into the CC and the CA, indicating that ILPs can reach the CA via local circuits and the terminal varicosities found in both the CC and the CA also suggest the release of ILPs into the hemolymph. Although not studied here, JH production could also be modulated by amino acids (AAs). Circulation is an important factor by which the CA may be called into action following a blood meal [65]. It is possible that AAs and other nutrients reach the CA through the dorsal vessel after feeding, increasing the activity of the CA and thereby favoring egg production. In *A. aegypti* and the brown planthopper, *Nilaparvata lugens*, AAs promote JH production [4,61]; indeed, in *N. lugens, JHAMT* expression is regulated by AA signaling [61]. Thus, by activating insulin signaling and/or by sensing AAs or other nutrients, the CA of *R. prolixus* synthesize and release JHSB_3_ at high levels after a blood meal. Interestingly, when ex vivo experiments are performed using the CC-CA complex attached to the brain, JH levels are higher in the incubation media with respect to those containing only the CC-CA complex. The brain is the main source of allatoregulators, and so in ex vivo assays, it is possible that the brain is exerting an influence over JH production via these regulators. 

In *B. mori,* injection of glucose promotes the release of an ILP from the brain [66]. Additionally, in *R. prolixus*, glucose injection leads to an increase in the insulin signaling pathway in the fat body, demonstrating that insulin is likely responsive to hemolymph glucose fluctuations [67]. Therefore, this carbohydrate appears to be a common nutritional signal for inducing the release of mammalian and insect insulins. Here, we demonstrate a direct effect of glucose potentially promoting insulin release from the brain MNSCs and thereby modulating JH synthesis in the CA. In addition, glucose is metabolized by the glycolytic pathway to produce citrate, a metabolite that can be broken down to acetyl-CoA, thus providing the immediate carbon source to fuel JH synthesis in the CA, as suggested in mosquitoes [68]. 

### 3.3. JHSB_3_ Biosynthesis Is Regulated by Allatoregulatory Neuropeptides and Possibly by Negative Feedback

JH production is also regulated by allatoregulatory neuropeptides; AT was the first neuropeptide described with a stimulatory effect on JH synthesis and release [69]. In *R. prolixus*, AT causes contractions of salivary glands and salivary secretion [70]. However, AT-like immunoreactive neurons in the brain have axonal projections exiting the brain and arborizing over the CC-CA complex. Here, for the first time in *R. prolixus*, we demonstrate the functional role of AT on the CA, positively modulating JH synthesis; accordingly, *AT-R* mRNA is upregulated in the CC-CA complex during vitellogenesis, where higher JH titers are detected. Interestingly, in the silkworm *Bombyx mori*, AT-R is mainly expressed in the short neuropeptide F (sNPF)-producing cells in the CC and not in the CA. AT could be inhibiting the allatostatic effect of sNPFs produced in the CC, thereby exerting an indirect allatotropic effect [71]. In *R. prolixus,* sNPF seems to promote production and laying of more eggs [72], therefore sNPF does not appear to be an inhibitor of reproduction in this insect. ASTs are pleiotropic neuropeptides widely distributed in insects and crustaceans and other invertebrates but not in vertebrates, for which one function in insects is the inhibition of juvenile hormone synthesis in a rapid and reversible way [73]. Neurons expressing FGLa/AST have been mapped in the brain of *R. prolixus,* with their axons projecting to the CC and around the CA, but no projections were reported within the CC-CA complex [74]. In *R. prolixus*, AST-2, an FGLa/AST, has been suggested to inhibit oogenesis [72]. Here, our results suggest that AST-2 inhibits JHSB_3_ synthesis in the isolated CC-CA. In addition, ASTC also plays an allatostatic role on JH production in several holometabola [68,75]. However, in *R. prolixus*, ASTC has been only studied in the context of myo- and cardio-regulation [76].

Recently, we reported that JH signaling, through the Met-Tai complex, promotes a negative feedback loop to maintain control over JHSB_3_ biosynthesis [51], a mechanism also suggested in the bug *Pyrrhocoris apterus* and in *S. gregaria* [77,78]. Here, when JHSB_3_ is topically applied to *R. prolixus* females, mRNA levels of two of the most important biosynthetic enzymes, *JHAMT* and *Epox*, are significantly reduced, thereby modulating JHSB_3_ production. Thus, we confirm the negative feedback loop between JHSB_3_ production and circulating JHSB_3_. Interestingly, Epox is the enzyme that shows the highest changes, again pointing out the importance of the epoxidation.

### 3.4. Insulin Signaling Regulates Ecdysteroid Production in the Ovaries

In many insect species, the adult ovaries are a main source of ecdysteroids, which play crucial roles in female reproduction [2,5]. Ecdysteroids in *R. prolixus* adult females promote the release of a myotropin ovulation/oviposition neurohormone from the MNSCs in the brain [41,47]; however, an additional role has recently been found, with ecdysteroids regulating ovarian growth and oocyte maturation [52]. The hemolymph of unfed *R. prolixus* adult females contains negligible ecdysteroid levels, but interestingly, these titers increase immediately and significantly after a blood meal [79], as do the transcripts for their biosynthetic enzymes [52]. In *R. prolixus*, follicular cells express Rhopr-IRs that activate the classical insulin pathway cascade after a blood meal [16,17]. Here, we describe that insulin not only increases transcript expression of the ecdysteroid biosynthetic enzymes (*Halloween genes*), but also stimulates ecdysteroid synthesis and release. In *A. aegypti,* a stimulatory effect of insulin on ecdysteroid production by ovaries, controlled by the tyrosine kinase activity of the insulin receptor, has been demonstrated [20,80]. In another dipteran, the blowfly *Phormia regina*, ILPs from the MNSCs progressively stimulate steroidogenesis in ovaries via its well-conserved transduction cascade [81]. Interestingly, because of the daily rhythm of ecdysteroids and ILP release from the ovaries and brain, respectively [61], Cardinal-Aucoin et al. [79] already suggested that ILPs may be involved in the regulation of ovarian ecdysteroid synthesis in *R. prolixus*. 

Collectively, the data presented here unmask endocrine communications of different factors that are critical for successful reproduction in *R. prolixus* females. Considering the literature already reported, and taking into account the results presented here, a proposed schematic model for the crosstalk of nutrition, insulin, juvenile hormone, and ecdysteroid signaling in the regulation of physiological processes required for successful reproduction is shown in Figure 8. Given the central role of the CA in insect reproduction, sophisticated and precise mechanisms of activation and inhibition have evolved to ensure both timely production and cessation. After a blood meal, ILPs are released from the brain and reach the CA to stimulate the expression of JH biosynthetic enzymes and thus, JHSB_3_ production. Because JH is not stored in the CA, JHSB_3_ is released into the circulation after its synthesis. In a coordinated manner, regulation of JH production and release is also mediated by two important classes of neuropeptides, stimulating (ATs) or inhibiting (ASTs). Once circulating, JHSB_3_ acts as a JH sensor through a negative feedback loop, directly modulating the transcription of JH biosynthetic enzymes to keep JHSB_3_ levels under control. A potential role of AA signaling and glucose on JH production is also proposed in this model. Insulin is also able to stimulate ecdysteroid synthesis in the ovary, increasing transcript expression of *Halloween genes*, and also ecdysteroid release into the circulation. Taken together, these coordinated actions of the neuro/endocrine system lead to successful reproduction. Our work provides a firm foundation upon which such further analyses can be based.

## 4. Materials and Methods

### 4.1. Insects 

Insects were taken from an established colony maintained at 25 °C and 50% humidity at the University of Toronto Mississauga. Fifth instar males and females (identified by the characteristic ventral midline crease in the cuticle of the segment surrounding the anus in females) were separated and then artificially fed 30 days post-ecdysis from fourth instars [82]. Insects that gorged at least nine times their own initial body weight were chosen and allowed to molt into adults. All the experiments were performed on mated insect females. Therefore, newly-emerged adult females were segregated individually and placed together with two recently fed males [83]. Mating was verified by examining the cubicle for the presence of the spermatophore [83]. After copulation, females were separated in two groups, one of them was left without feeding and the another was fed on blood 10 days post-ecdysis to promote egg growth. Only adult females that fed 2.5 to 3 times their initial body weight were used in the experiments [16]. Brains, CC-CA complexes, fat bodies or ovaries, were dissected in cold autoclaved phosphate-buffered saline (PBS, 8.2 mM Na_2_HPO_4_, 1.5 mM KH_2_PO_4_, 150 mM NaCl, 2.7 mM KCl) on representative days of the unfed and fed condition, as indicated. Hemolymph was obtained from adult females by cutting the legs and pressing gently the abdomen. For JHSB_3_ quantification, hemolymph was collected in silanized vials (Thermo Fisher Scientific, Waltham, MA, USA) containing anticoagulant solution (PBS with 10 mM EDTA, 26 mM sodium citrate, 26 mM citric acid and 100 mM glucose). 

### 4.2. RNA Extraction and Reverse Transcription/Quantitative PCR (RT-qPCR)

Total RNA was extracted from tissues using TRIzol reagent (Invitrogen by Thermo Fisher Scientific, Waltham, MA, USA) according to the manufacturer’s instructions. Synthesis of first-strand complementary DNA (cDNA) was done using the Applied Biosystems High Capacity cDNA Reverse Transcription Kit (Applied-Biosystems, by Thermo Fisher Scientific, Mississauga, ON, Canada). cDNA final concentration and its A260/280 ratio were measured using a spectrophotometer DS-11+ (DeNovix Inc., Wilmington, DE, USA). Total RNA extracts were then subjected to DNase treatment using DNase I (RNase-free) Kit (Thermo Fisher Scientific, Mississauga, ON, Canada) [51]. Quantitative PCR was performed using an advanced qPCR 1-Step Kit with Supergreen Dye Low-ROX (Wisent Bioproducts Inc., Saint-Jean-Baptiste, QC, Canada). *β-actin* and *ribosomal protein 49* (Rp49) were used as reference genes. Primers are shown in Appendix A. For each pair of primers, the efficiency ranged from 88 to 104%, with linear correlation coefficients (r^2^) ranging from 0.8 to 1, and the dissociation curves always showed a single peak. mRNA expression was calculated relative to 1000 copies of the average of the reference genes using 2^−ΔCt^ method [84] or as fold change quantified relative to the expression of control samples, using 2^−ΔΔCt^ method following of the geometric mean of the reference genes [85]. Experiments were repeated with at least five biological replicates, as indicated in each experiment, having two technical replicates and using no-template controls.

### 4.3. In Vivo Experiments: JHSB_3_ and Insulin Treatments

Newly emerged adult females were topically treated on the cuticle of the dorsal abdomen with either 10 μL of acetone containing 50 pg of JHSB_3_ (Toronto Research Chemical, North York, ON, Canada), or 10 μL of acetone (controls) [51]. The CC-CA complexes were removed 24 h after and processed for RT-qPCR. Another group of newly emerged adult females were injected into the hemocoel with 5 μL of 0.1 μg/μL porcine insulin (Millipore-Sigma, Oakville, ON, Canada) diluted in saline (150 mM NaCl, 8.6 mM KCl, 2 mM CaCl_2_, 4 mM NaHCO_3_, 34 mM glucose, 8.5 mM MgCl_2_, 5 mM HEPES [pH 7.2]) or 5 μL of saline (control). The CC-CA complexes, hemolymph, fat bodies and ovaries were removed 24 h after and processed for RT-qPCR or hormone quantification. Unfed females are preferred for these experiments because JH and ecdysteroid levels are low and insects are able to respond to insulin levels by rapidly activating insulin signaling [16].

### 4.4. Ex Vivo Experiments: JHSB_3,_ Insulin and Allatoregulator Treatments

Isolated CC-CA complexes from unfed females were incubated with JHSB_3_ in glass silanized vials, using 200 μL of Grace’s medium, with L-glutamine and without insect hemolymph (Millipore-Sigma, Oakville, ON, Canada) [51]. JHSB_3_ was dissolved in acetone and added to the incubation medium at a final concentration of 35 nM; controls were incubated with the same percentage of solvent alone. The CC-CA complexes were collected after 4 h of incubation (in the dark, at 28 °C and with gentle shaking) and processed for RT-qPCR. Tissues were also incubated for 4 h with different peptides in glass silanized vials using the tissue culture media M−199 containing 2% Ficoll, 25 mM HEPES (pH 6.5) and methionine (50 μM) [86], and then analyzed for JHSB_3_ quantification (final volume: 120 µL). The incubations were performed as follow: (a) CC-CA complexes from unfed females incubated with porcine insulin (final concentration: 3 μM); (b) CC-CA complexes pre-incubated for 1 h with LY294002 (Biogems, Westlake Village, CA, USA) (final concentration: 50 μM) and then with porcine insulin (final concentration: 3 μM); (c) CC-CA complexes attached to the brain from unfed females incubated with glucose (final concentration: 50 mM); (d) CA-CC complexes from fed females (3 days after feeding, when JHSB_3_ titers remain higher) incubated in the presence of Rhopr-AST-2 (LPVYNFGLamide, synthesized by GenScript, Piscataway, NJ, USA) at a final concentration of 10^−7^ M [87]; (e) CC-CA complexes from unfed females (10–12 days post ecdysis to adult, when JHSB_3_ titers remain lower), incubated with Rhopr-AT (NVQLSTARGFamide, synthesized by GenScript, Piscataway, NJ, USA) at a final concentration of 10^−7^ M [70]. Controls were not treated with any peptide. 

In addition, the ovaries from unfed females were collected and treated with porcine insulin (final concentration: 3 μM) or saline (control) using sterilized microtubes and of Grace’s medium (200 μL), as described above. The tissues and the incubation medium were collected after 4 h of incubation (in the dark, at 28 °C and with gentle shaking) and processed for RT-qPCR and ecdysteroid quantification, as detailed below. 

### 4.5. JHSB_3_ Identification and Quantification 

Recently, the screening for five JH homologs in the hemolymph of 4th instar nymphs of *R. prolixus* revealed only the presence of JHSB_3_ hemolymph. In order to corroborate if in adult females any other JH homolog is present, JH homolog screening was performed using a liquid chromatography coupled to tandem mass spectrometry protocol, as previously described by Ramirez et al. [88]. The JHSB_3_ amounts present in the hemolymph were quantified using a deuterated JH III analog (JH III-D3, from Toronto Research Chemicals, North York, ON, Canada) as an internal standard, detailed by Villalobos-Sambucaro et al. [50]. Briefly, 70 μL of hemolymph or incubation medium were collected for each insect and placed in cold glass silanized vials (Thermofisher Scientific, Waltham, MA, USA) containing 60 μL of anticoagulant solution. After that, 10 μL of 6.25 ppb of JH III-D3 in acetonitrile were added to each sample, followed by 600 μL of hexane. Samples were vortexed for 1 min, and centrifugated for 5 min at 4 °C and 2000× *g*. The organic phase was transferred to a new silanized vial, dried under nitrogen flow, and stored at −20 °C. Dried extracts were re-suspended in 50 μL of acetonitrile, vortexed 1 min, transferred to a new silanized vial with a fused 250 μL insert. The identification and quantification of JHSB_3_ in the hemolymph on representative days of the reproductive cycle of *R. prolixus* was based on multiple reaction monitoring (MRM), using the two most abundant fragmentation transitions: 283 → 233 (primary) and 283 → 145 (secondary) [88]. 

### 4.6. Quantification of Vitellogenin by ELISA

Quantification of Vg in the hemolymph was carried out by the enzyme-linked immunosorbent assay (ELISA) as described by Leyria et al. [51]. Briefly, microtiter plates were loaded with two technical replicates using 200 μL/well of standard Vg or with appropriate hemolymph dilutions in buffer carbonate (15 mM Na_2_CO_3_, 35 mM NaHCO_3_, pH 9.6) and incubated for 90 min at 37 °C. After washing, plates were incubated with anti-Vg antibody (0.01 µg/mL) for 60 min at 37 °C. Plates were then loaded with anti-rabbit immunoglobulin conjugated to horseradish peroxidase (HRP) in PBST (1:5000) for 30 min at 37 °C. Finally, plates were incubated with TMB Liquid Substrate System (Millipore-Sigma, Oakville, ON, Canada) for 15 min and then stopped with 4 N H_2_SO_4_. Plates were read at 492 nm using a multi-mode reader (Synergy HTX from Agilent Technologies, Santa Clara, CA, USA). A standard curve using the recombinant antigen sequence (from 50 pg/mL to 10 mg/mL) was performed in parallel. 

### 4.7. Ecdysteroid Quantification 

Hemolymph from insulin-treated insects or incubation medium from ex vivo assays were collected to measure ecdysteroid titers. The samples were combined with methanol at a ratio (1:3) (sample:methanol), and then stored at −20 °C. Competitive ELISA, using 20E conjugated to HRP reagent and a rabbit anti-ecdysteroid primary antibody, was used to quantify hemolymph ecdysteroid titers. Details of the procedure are described by Abuhagr et al. [89] and Benrabaa et al. [52]. 

### 4.8. Statistical Analyses

All data were processed using the GraphPad Prism 9 Software (GraphPad Software, San Diego, CA, USA). All datasets passed normality and homoscedasticity tests. Significance of differences were determined either with one-tailed Student’s *t*-test, or with one-way ANOVA followed by Tukey’s test, as indicated.

## Figures and Tables

**Figure 1 ijms-24-00007-f001:**
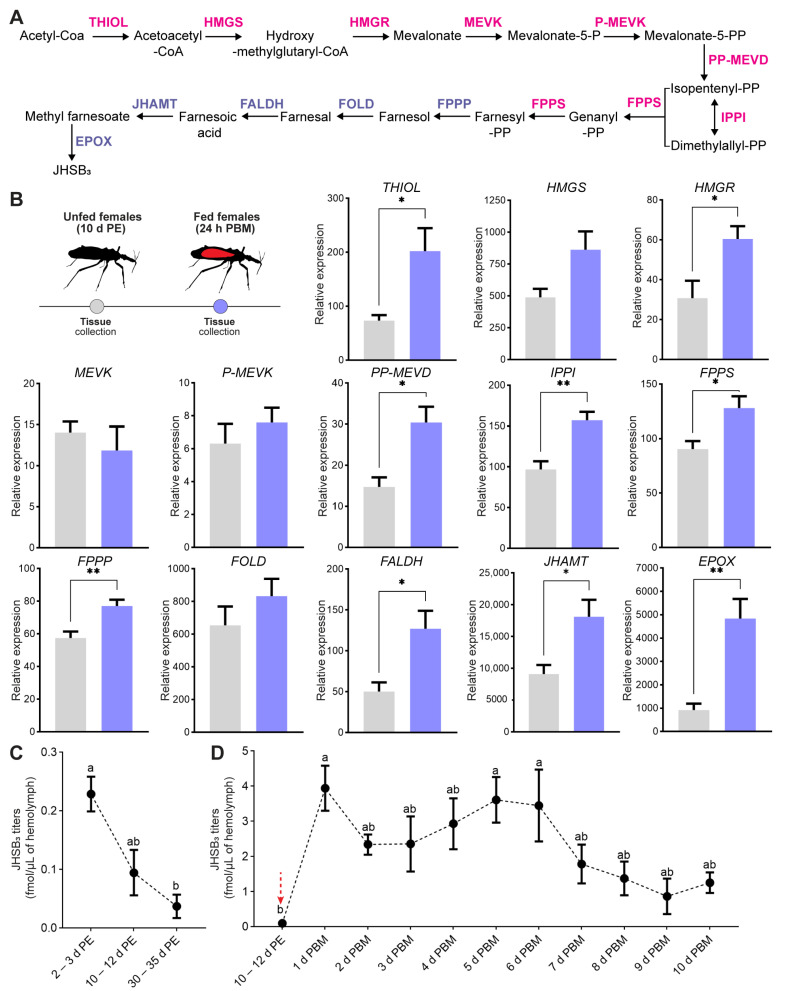
JHSB_3_ production in *R. prolixus* after a blood meal. (**A**) Scheme of JHSB_3_ biosynthesis. Precursors are connected by arrows; enzymes of the early step are in pink and enzymes of the late step in lilac. (**B**) Transcript levels of JH biosynthetic enzymes were quantified in the CC-CA complexes from unfed (10 d PE) and fed (24 h PBM) females using RT-qPCR. The analysis was performed using the 2^−ΔCt^ method. The y axes represent the relative expression obtained via geometric averaging using *Rp49* and *actin* as reference genes. The results are shown as the mean ± SEM (*n* = 5, where each *n* represents a pool of CC-CA complexes from 3 insects). ** *p* <* 0.05; *** *p* <* 0.01 (Student’s *t*-test). (**C**,**D**) Hemolymph was collected from adult females at different time points post ecdysis (**C**), and throughout 10 days after the first blood meal as an adult (**D**). The y axes represent JHSB_3_ titers (fmol) per µL of hemolymph (black symbols). The results are shown as mean ± SEM of *n* = 6–7 (each *n* is hemolymph from 5–7 insects). Red arrow: time point chosen to feed insects. Different letters indicate significant differences at p < 0.05 (one-way ANOVA and Tukey’s test as post-hoc test). d PE, days post ecdysis; h PBM, hours post blood meal.

**Figure 2 ijms-24-00007-f002:**
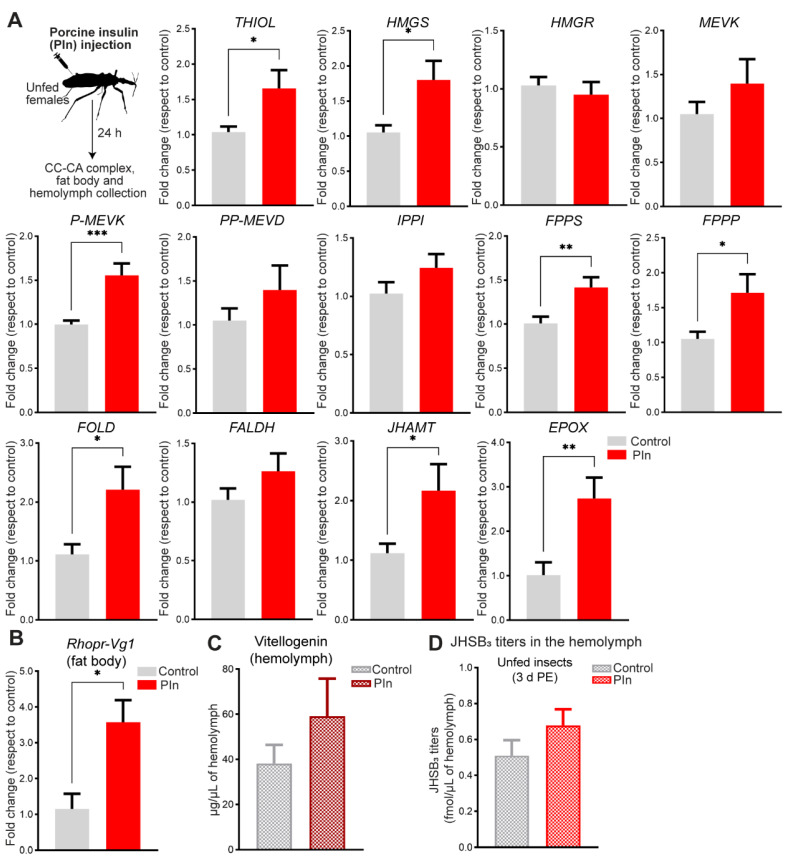
Role of insulin in JHSB_3_ production in in vivo assays. (**A**) Unfed adult females were injected with porcine insulin (PIn, 0.5 µg) and 24 h later, the CC-CA complexes were dissected to measure transcript expression of the JH biosynthetic enzymes. Transcript levels were quantified using RT-qPCR and analyzed by the 2^−ΔΔCt^ method. The y axes represent the fold change in expression relative to control (saline injection, gray column, value ~1) obtained via geometric averaging using *Rp49* and *actin* as reference genes. The results are shown as the mean ± SEM (*n* = 5, where each *n* represents a pool of CC-CA complexes from 3 insects). ** *p* <* 0.05; *** *p* <* 0.01; **** p* < 0.001 (Student’s *t*-test). (**B**) Effect of insulin injection on *Rhopr-Vg1* mRNA expression. Transcript levels were quantified in the fat body using RT-qPCR and analyzed by the 2^−ΔΔCt^ method. The y axis represents the fold change in expression relative to control (saline injection, gray column, value ~1) obtained via geometric averaging using *Rp49* and *actin* as reference genes. The results are shown as the mean ± SEM of *n* = 5, where each *n* represents a pool of fat bodies from 3 insects. ** *p* <* 0.05 (Student’s *t*-test). (**C**) Quantification of vitellogenin in the hemolymph of unfed insulin-injected females by ELISA. The results are shown as the mean ± SEM (*n* = 5–6, where each *n* represents hemolymph from 1 insect). (**D**) Hemolymph was collected from unfed adult females (3 days post ecdysis) 24 h after insulin injection. The y axes represent JHSB_3_ titers (fmol) per µL of hemolymph. The results are shown as mean ± SEM of *n* = 4–5, where each *n* is hemolymph from 5–7 insects. d PE, days post ecdysis.

**Figure 3 ijms-24-00007-f003:**
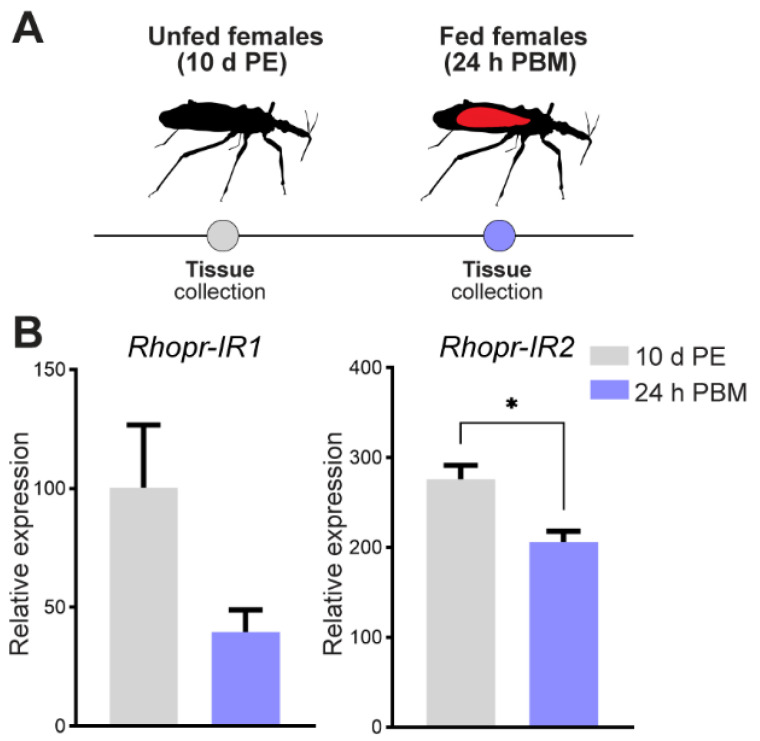
Transcript expression of insulin receptors in the CC-CA of *R. prolixus* in different nutritional states. (**A**) Scheme of tissue collection. (**B**) Transcript levels of *Rhopr-IR1* (**left**) and *Rhopr-IR2* (**right**) were quantified in the CC-CA complexes obtained from unfed (10 d PE) and fed (24 h PBM) females using RT-qPCR. The analysis was performed using the 2^−ΔCt^ method. The y-axes represent the relative expression obtained via geometric averaging using *Rp49* and *actin* as reference genes. The results are shown as the mean ± SEM (*n* = 5, where each *n* represents a pool of CC-CA complexes from 3 insects). * *p* < 0.05; (Student’s *t*-test). d PE, days post ecdysis; h PBM, hours post blood meal.

**Figure 4 ijms-24-00007-f004:**
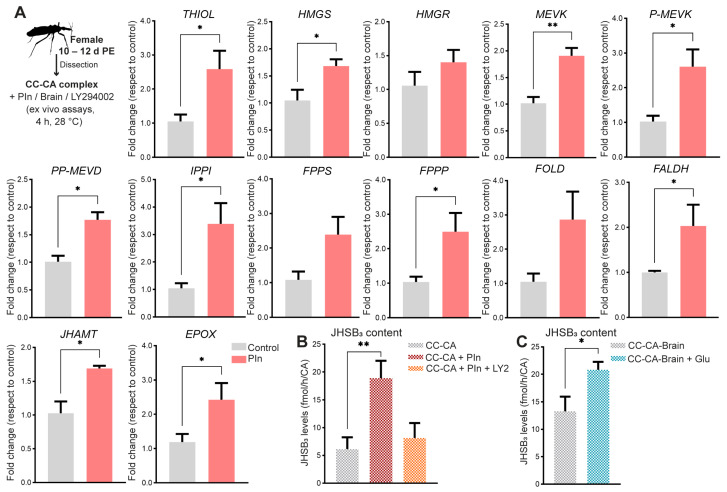
Role of insulin in JHSB_3_ production in ex vivo assays. (**A**) CA-CC complexes from unfed females (10–12 d post-ecdysis) were incubated ex vivo at 28 °C in the dark, in culture medium supplemented with porcine insulin (3 μM), or the same volume of saline (control). Four hours later, transcript expression of the JH biosynthetic enzymes in the CC-CA complex were quantified using RT-qPCR and analyzed by the 2^−ΔΔCt^ method. The y axes represent the fold change in expression relative to control (saline, gray column, value ~1) obtained via geometric averaging using *Rp49* and *actin* as reference genes. The results are shown as the mean ± SEM of *n* = 5, where each *n* represents a pool of CC-CA complexes from 3 insects. ** *p* <* 0.05; *** *p* <* 0.01; (Student’s *t*-test). (**B**) JHSB_3_ levels in culture media from: group 1, the CC-CA complexes alone; group 2, CC-CA complexes incubated with porcine insulin (3 μM); and group 3, the CC-CA complexes pre-incubated for 1 h with LY294002 (Ly2, final concentration: 50 μM) prior to porcine insulin (3 μM) treatment. The y axes represent JHSB_3_ titers (fmol) per h per CA. The results are shown as the mean ± SEM of *n* = 5–7, where each *n* represents the incubation medium that contained a pool of tissues from 3 insects. *** *p* <* 0.01; (Student’s *t*-test). (**C**) JHSB_3_ levels in culture media from: group 1, CC-CA complexes connected to the brain (CC-CA-Brain); group 2, CC-CA-Brain incubated with glucose (Glu, 50 mM). The y axes represent JHSB_3_ titers (fmol) per h per CA. The results are shown as the mean ± SEM of *n* = 3–4, where each *n* represents the incubation medium that contained a pool of tissues from 3 insects. ** *p* <* 0.05; (Student’s *t*-test).

**Figure 5 ijms-24-00007-f005:**
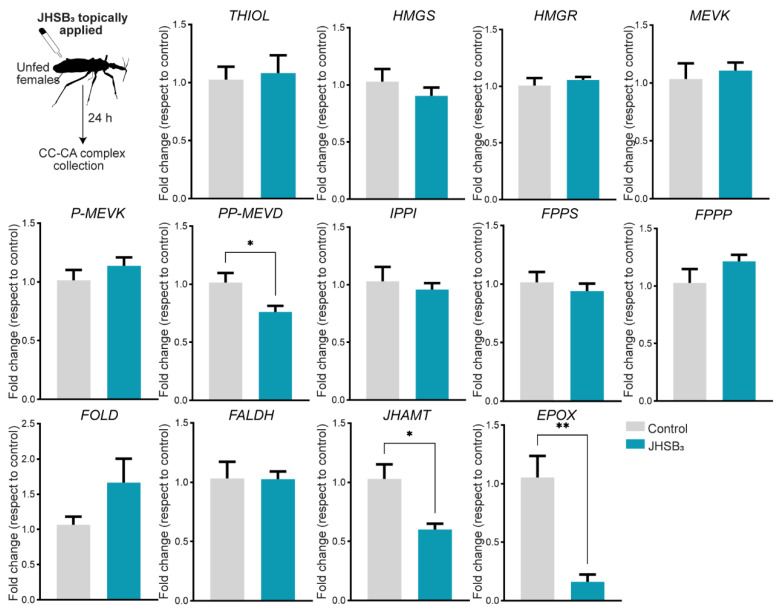
Effect of JHSB_3_ on its own synthesis. Newly emerged adult females were treated with exogenous JHSB_3_ (50 pg diluted in acetone), and 24 h later, transcript levels of JHSB*_3_* biosynthetic enzymes were analyzed using RT-qPCR and analyzed by the 2^−ΔΔCt^ method. The y axes represent the fold change in expression relative to control (insect treated with acetone, gray column, value ~1) obtained via geometric averaging using *Rp49* and *actin* as reference genes. The results are shown as the mean ± SEM of *n* = 5, where each *n* represents a pool of CC-CA complexes from 3 insects. ** *p* <* 0.05; *** *p* <* 0.01; (Student’s *t*-test).

**Figure 6 ijms-24-00007-f006:**
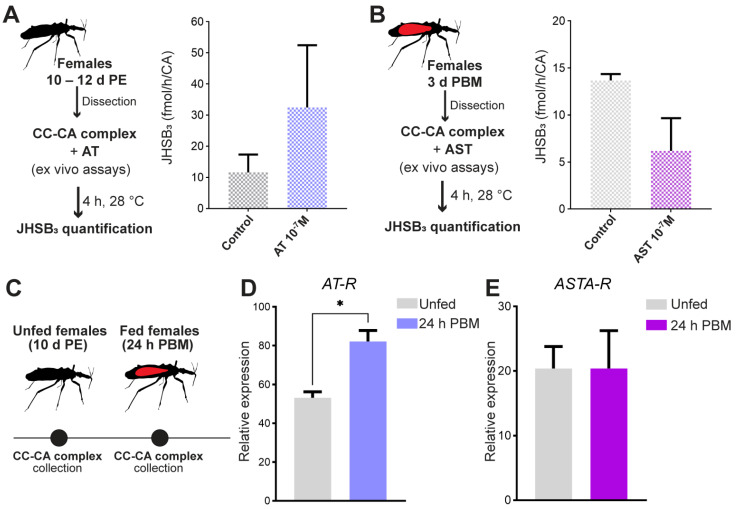
Effect of AT and AST on JHSB_3_ synthesis in ex vivo assays. (**A**) CA-CC complexes from unfed females (10–12 d PE), were incubated ex vivo at 28 °C in the dark, for 4 h, in culture medium supplemented with AT (final concentration: 10^−7^ M) or the same volume of saline (control). The y axes represent JHSB_3_ levels with respect to control (saline, gray column). The y axis represents JHSB_3_ titers (fmol) per h and per CA (*n* = 3, where each *n* represents a pool of CC-CA complexes from 3 insects). (**B**) CA-CC complexes from fed females (3 d PBM) were incubated ex vivo at 28 °C in the dark, for 4 h, in culture medium supplemented with AST-2 (final concentration: 10^−7^ M) or the same volume of saline (control). The y axis represents JHSB_3_ titers (fmol) per h and per CA (*n* = 3–4, where each *n* represents a pool of CC-CA complexes from 3 insects). (**C**) Scheme of transcript expression analysis for *AT receptor* (AT-R) and *AST receptor* (ASTA-R) in *R. prolixus* prior to and post blood meal. (**D**,**E**) Transcript levels of *AT-R* (**D**) and *ASTA-R* (**E**) were quantified in the CC-CA complexes using RT-qPCR. The analysis was performed using the 2^−ΔCt^ method. The y axes represent the relative expression obtained via geometric averaging using *Rp49* and *actin* as reference genes. The results are shown as the mean ± SEM (*n* = 5–6, where each *n* represents a pool of CC-CA complexes from 3 insects). ** *p* <* 0.05 (Student’s *t*-test). d PE, days post ecdysis; d PBM, days post blood meal.

**Figure 7 ijms-24-00007-f007:**
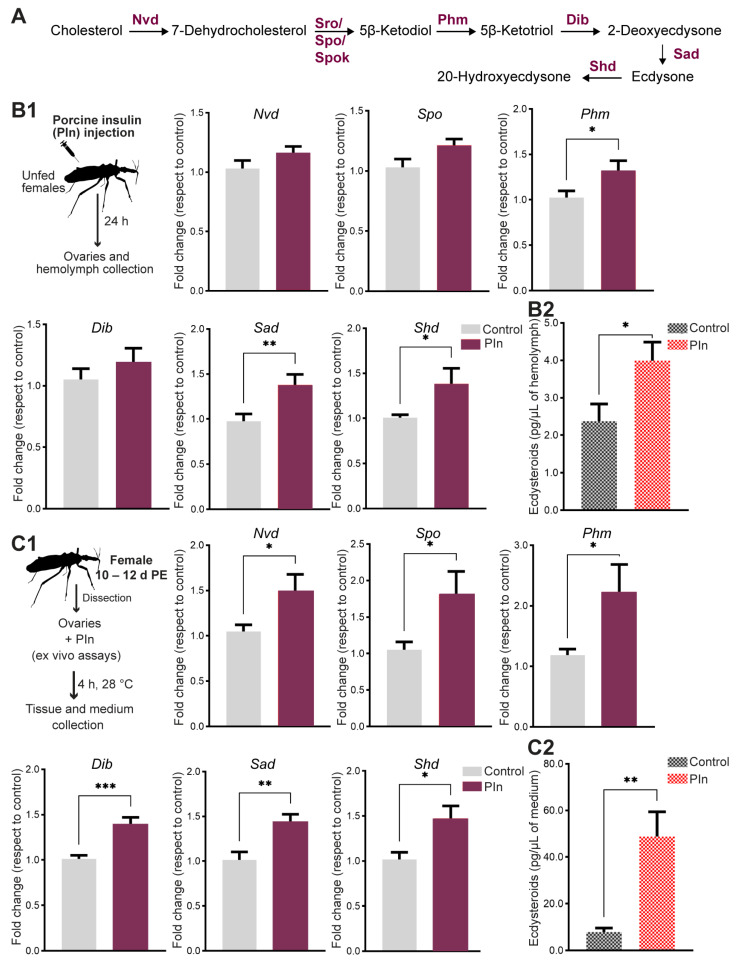
Role of insulin in ecdysteroid production. (**A**) Scheme of ecdysteroid biosynthesis. (**B1**) Unfed adult females were injected with porcine insulin (PIn, 0.5 µg diluted in saline) and 24 h later the ovaries were dissected to measure transcript expression of ecdysteroid biosynthetic enzymes. Transcript levels were quantified using RT-qPCR and analyzed by the 2^−ΔΔCt^ method. The y axes represent the fold change in expression relative to control (saline injection, gray column, value ~1) obtained via geometric averaging using *Rp49* and *actin* as reference genes. The results are shown as the mean ± SEM (*n* = 7–8, where each *n* represents ovaries from 1 insects). ** *p* <* 0.05; *** *p* <* 0.01 (Student’s *t*-test). (**B2**) Ecdysteroid hemolymph titers after insulin injection were quantified by ELISA. The y axes represent ecdysteroid levels (pg) per µL of hemolymph. The results are shown as the mean ± SEM (*n* = 8, where each *n* represents hemolymph from 1 insects). ** *p* <* 0.05 (Student’s *t*-test). (**C1**) Ovaries from unfed females (3 d post-ecdysis) were incubated ex vivo at 28 °C in the dark, in Grace’s insect medium. The culture medium was supplemented with porcine insulin (3 μM), or the same volume of saline (control). Four hours later, transcript expression of the ecdysteroid biosynthetic enzymes were quantified using RT-qPCR and analyzed by the 2^−ΔΔCt^ method. The y-axes represent the fold change in expression relative to control (saline, gray column, value ~1) obtained via geometric averaging using *Rp49* and *actin* as reference genes. The results are shown as the mean ± SEM of *n* = 5–6, where each *n* represents ovaries from 1 insect. ** *p* <* 0.05; *** *p* <* 0.01; **** p* < 0.001 (Student’s *t*-test). (**C2**) Ecdysteroid hemolymph levels in the incubation medium were quantified by ELISA. The y axes represent ecdysteroid levels (pg) per µL of medium. The results are shown as the mean ± SEM (*n* = 4–5, where each *n* represents the incubation medium that contained the ovaries of 1 insect. *** *p* <* 0.01 (Student’s *t*-test).

**Figure 8 ijms-24-00007-f008:**
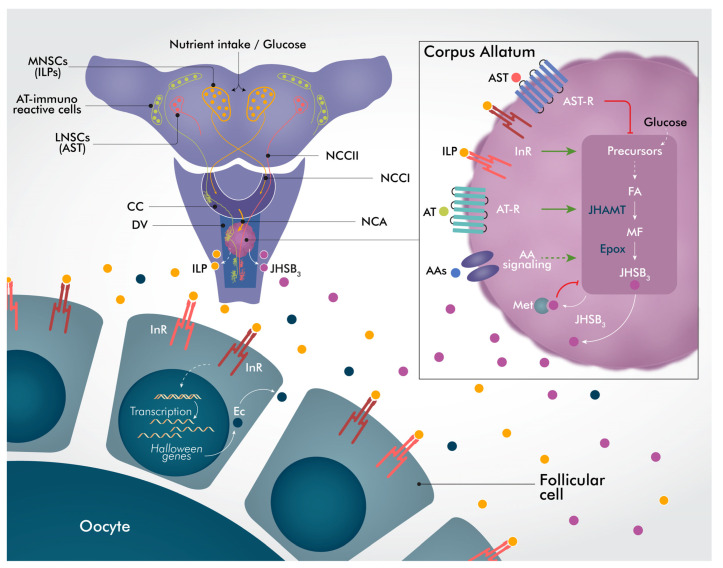
Insights into the crosstalk between nutrition, insulin, juvenile hormone, and ecdysteroid signaling for successful reproduction in *R. prolixus* females. After a blood meal, ILPs stored in the MNSCs are released from processes in the CC and CA. In the CA, ILPs stimulate expression of JH biosynthetic enzymes, and thus JHSB_3_ production. JH production and release is also likely regulated by two important classes of neuropeptides. Neurosecretory cells containing ATs and others containing ASTs have axons that exit the brain via the NCCII delivering peptides that can stimulate (ATs) or inhibit (ASTs) CA activity. Once circulating, JHSB_3_ acts via Met as a JH sensor through a negative feedback loop, directly modulating the transcription of its own biosynthetic enzymes. A potential role of AA signaling to stimulate JH production after a blood meal is also proposed. Moreover, glucose might modulate JH production by stimulating the release of ILPs from the brain (ILPs might also influence AT/AST release) or working as a source of precursors for JH synthesis. Interestingly, JHAMT and Epox appear to be the enzymes most regulated. Insulin, in turn, stimulates ecdysteroid synthesis in the ovary by increasing transcript expression of *Halloween genes*, and ecdysteroid release into the hemolymph. Together, these coordinated actions lead to a successful reproductive cycle. JHSB_3_, JH III skipped bisepoxide; AST, an FGLa/allatostatin type A; AT, allatotropin; JHAMT, Juvenile hormone acid methyltransferase; Epox, Methyl farneseoate epoxidase; ILPs, insulin-like peptides; AT-R, allatotropin GPCR; AST-R, allatostatin GPCR; InR, insulin receptors; CC, corpus cardiacum; NCCI, nervus corpus cardiacum I; NCCII, nervus corpus cardiacum II; NCA, nervus corpus allatum; LNSC, lateral neurosecretory cell; MNSCs, median neurosecretory cells; Ec, ecdysteroids; DV, dorsal vessel.

## Data Availability

The data that support the findings of this study are available in the main text and Appendix A of this article.

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
