# Peer review of "Crosstalk between Nutrition, Insulin, Juvenile Hormone, and Ecdysteroid Signaling in the Classical Insect Model, Rhodnius prolixus"

_ijms, 2022, doi:10.3390/ijms24010007_

Round 1

Reviewer 1 Report

This study addresses the complexity of insect molecular physiological systems, by an analysis of feeding effects on specific hormones relevant for development and reproduction. The work uses Rhodnius prolixus, an insect species relevant as physiological model and as an important vector species for Chagas disease. The level of analysis is at transcript levels and hormone JHSB3 levels, studied by feeding experiments, hormone treatments, and ex vivo experiments. The work fits into the framework of International Journal of Molecular Sciences, and the documentation of the experiments is adequate and uses very clear figures to illustrate the findings. The experiments find connections of feeding and insulin signals on Juvenile hormone, including a self-regulation, and on ecdysteroid hormone pathway.

The experiments are carefully set up and combine different approaches to substantiate the involvement of insulin signalling on insect hormone levels. The presentation is fairly compact, but the figures illustrate the findings well. The authors may consider to introduce sections into the discussion to break down the different steps presented in the results. Overall, I have only minor comments on this manuscript. 

L 24     two of the main lipophilic hormones (see also line 73) – since ecdysteroids are presented rather as a class of hormones then a single hormone, probably rephrase as “two main types of hormones”, or include the names of the ecdysteroids analysed here

L 26     defined – probably better: established?

L 27     ex vivo experiments – probably explain on which organs/ tissues were used

L 44     the reference included here covers JH, consider additional references for other hormones which could be relevant as well

L 59 – 67        this section addresses the diversity of ILP signalling in insects, but could also mention the situation in Rhodnius

L 94     include a reference to Figure 1 here?

L 124   The end of the introduction reads rather general. Consider to outline the following experiments, to guide the reader through the different approaches and different findings that follow.

L 135   Should also discuss why the transcript level for enzymes is not seen in all enzymes, but for the majority.

Fig. 2a The drop in transcript levels in HMGR sticks out in this panel, is there a reason for the decrease? It contrasts with the increase upon feeding in Fig. 1b

L 459   the CA thor-ough à through?

L 691   Abbreviations are not listed alphabetically, which is not helpful for orientation when looking up a specific abbreviation. Abbreviation for Acetyl-CoA-thiolase is lacking.

L 794   The reference lacks a volume

L 816   reference lacks a comma between year and volume,

See also lines 823, 780, 788

L 873   set species name in italics in this line

Author Response

We thank the reviewers for the positive and constructive comments on the MS. Below we address each of the concerns raised by reviewers in detail; our point-by-point responses are highlighted in red.

Reviewer 1

This study addresses the complexity of insect molecular physiological systems, by an analysis of feeding effects on specific hormones relevant for development and reproduction. The work uses Rhodnius prolixus, an insect species relevant as physiological model and as an important vector species for Chagas disease. The level of analysis is at transcript levels and hormone JHSB3 levels, studied by feeding experiments, hormone treatments, and ex vivo experiments. The work fits into the framework of International Journal of Molecular Sciences, and the documentation of the experiments is adequate and uses very clear figures to illustrate the findings. The experiments find connections of feeding and insulin signals on Juvenile hormone, including a self regulation, and on ecdysteroid hormone pathway. The experiments are carefully set up and combine different approaches to substantiate the involvement of insulin signalling on insect hormone levels. The presentation is fairly compact, but the figures illustrate the findings well. The authors may consider to introduce sections into the discussion to break down the different steps presented in the results. Overall, I have only minor comments on this manuscript.

Response 1: We thank the Reviewer for this comment; as requested, we have inserted sections into the discussion for a better understanding.

L 24 two of the main lipophilic hormones (see also line 73) – since ecdysteroids are presented rather as a class of hormones then a single hormone, probably rephrase as “two main types of hormones”, or include the names of the ecdysteroids analysed here

Response 2: Now modified.

L 26 defined – probably better: established?

Response 3: Now modified.

L 27 ex vivo experiments – probably explain on which organs/ tissues were used

Response 4: Now included.

L 44 the reference included here covers JH, consider additional references for other hormones which could be relevant as well

Response 5: Now included.

L 59 – 67 this section addresses the diversity of ILP signalling in insects, but could also mention the situation in Rhodnius

Response 6: Now included.

L 94 include a reference to Figure 1 here?

Response 7: Now included as Figure 1A.

L 124 The end of the introduction reads rather general. Consider to outline the following experiments, to guide the reader through the different approaches and different findings that follow.

Response 8: Now included.

L 135 Should also discuss why the transcript level for enzymes is not seen in all

enzymes, but for the majority.

Response 9: As requested, we have now inserted a phrase about this result in Discussion.

Fig. 2a The drop in transcript levels in HMGR sticks out in this panel, is there a reason for the decrease? It contrasts with the increase upon feeding in Fig. 1b

Response 10: We don't think it's a significant drop in transcript levels. However, if so, it is probably due to the production of other compounds, such as defensive secretions and pheromones, which are also synthesized via the mevalonic acid (MVA) pathway. For example, HMGR is critical for the biosynthesis of a monoterpene defensive toxin in insects involved in chemical defence as well as in courtship and mating behaviours.

L 459 the CA thor-ough à through?

Response 11: Now corrected.

L 691 Abbreviations are not listed alphabetically, which is not helpful for orientation when looking up a specific abbreviation. Abbreviation for Acetyl-CoA-thiolase is lacking.

Response 12: Now corrected.

L 794 The reference lacks a volume

Response 13: Now added.

L 816 reference lacks a comma between year and volume, See also lines 823, 780, 788

Response 14: Now corrected.

L 873 set species name in italics in this line

Response 15: Now corrected.

Reviewer 2 Report

1.       My major concern is that, in the MS, the authors confirmed the transcript levels using RT-qPCR, and why the authors did not perform transcriptome sequencing to establish the digital gene expression?

2.       Line 551-552, how the authors identified the gender of Rhodnius prolixus during the nymphal instar?

3.       Line 577-578, among a series of r eference genes, how the authors selected β-actin and ribosomal protein 49 (Rp49) for normalizing the data?

Author Response

We thank the reviewers for the positive and constructive comments on the MS. Below we address each of the concerns raised by reviewers in detail; our point-by-point responses are highlighted in red.

Reviewer 2

My major concern is that, in the MS, the authors confirmed the transcript levels using RT-qPCR, and why the authors did not perform transcriptome sequencing to establish the digital gene expression?

Response 1: Thanks for the question. The aim of the work was to evaluate specific transcript expression of some genes of known sequence, so qPCR, as a highly sensitive and reliable tool to evaluate transcripts, was chosen in this work.

Line 551-552, how the authors identified the gender of Rhodnius prolixus during the nymphal instar?

Response 2: The sexes are easily identified in the fifth instar by the characteristic ventral midline crease in the cuticle of the segment surrounding the anus in females. We have inserted details on this. This was described many years ago but a recent drawing can be found in Chiang et al, 2013 (Insects, 4:593-608).

Line 577-578, among a series of reference genes, how the authors selected β-actin and ribosomal protein 49 (Rp49) for normalizing the data?

Response 3: We, and others, have in the past tested several housekeeping genes; β-actin and ribosomal protein 49 (Rp49) were selected as appropriate control genes for relative qPCR experiments because their expression levels were stable across the range of experimental conditions and treatments performed in this work.

Again, we appreciate the comments and corrections and believe the MS is all the better for it.

Best wishes,

Jimena Leyria, PhD